# Polymorphisms within the Tumor Necrosis Factor-Alpha Gene Is Associated with Preeclampsia in Taiwanese Han Populations

**DOI:** 10.3390/biomedicines11030862

**Published:** 2023-03-11

**Authors:** Chih-Wei Lin, Chung-Hwan Chen, Meng-Hsing Wu, Fong-Ming Chang, Lin Kang

**Affiliations:** 1Department of Obstetrics & Gynecology, College of Medicine, National Cheng Kung University, Tainan 70101, Taiwan; yowchi66@hotmail.com (C.-W.L.); mhwu68@mail.ncku.edu.tw (M.-H.W.); fchang@mail.ncku.edu.tw (F.-M.C.); 2Department of Obstetrics and Gynecology, National Cheng Kung University Hospital, College of Medicine, National Cheng Kung University, Tainan 70101, Taiwan; 3Orthopaedic Research Center, College of Medicine, Kaohsiung Medical University, Kaohsiung 80708, Taiwan; hwan@kmu.edu.tw; 4Department of Orthopedics, Kaohsiung Medical University Hospital, Kaohsiung Medical University, Kaohsiung 80708, Taiwan; 5Regeneration Medicine and Cell Therapy Research Center, Kaohsiung Medical University, Kaohsiung 80708, Taiwan; 6Departments of Orthopedics, College of Medicine, Kaohsiung Medical University, Kaohsiung 80708, Taiwan; 7Department of Orthopedics, Kaohsiung Municipal Ta-Tung Hospital, Kaohsiung Medical University, Kaohsiung 80145, Taiwan; 8Institute of Medical Science and Technology, National Sun Yat-Sen University, Kaohsiung 80420, Taiwan; 9Graduate Institute of Animal Vaccine Technology, College of Veterinary Medicine, National Pingtung University of Science and Technology, Pingtung 912301, Taiwan; 10Graduate Institute of Materials Engineering, College of Engineering, National Pingtung University of Science and Technology, Pingtung 912301, Taiwan; 11Ph.D. Program in Biomedical Engineering, College of Medicine, Kaohsiung Medical University, Kaohsiung 80145, Taiwan

**Keywords:** polymerase chain reaction (PCR), preeclampsia, rs1800629, rs361525, tumor necrosis factor-α (TNF-α)

## Abstract

Preeclampsia (PE) occurs in women pregnant for more than 20 weeks with de novo hypertension and proteinuria, and is a devastating disease in maternal–fetal medicine. Cytokine tumor necrosis factor (TNF)-α may play a key role in the pathogenesis of PE. We conducted this study to investigate the regulatory regions of the TNF genes, by investigating two promoter polymorphisms, TNFA-308G/A (rs1800629) and -238G/A (rs361525), known to influence TNF expression, and their relationship to PE. An observational, monocentric, case–control study was conducted. We retrospectively collected 74 cases of severe PE and 119 pregnant women without PE as control. Polymerase chain reaction (PCR) was carried out for allele analysis. Higher A allele in women with PE was found in rs1800629 but not rs361525. In this study, we first found that polymorphism at the position -308, but not -238, in the promoter region of the TNF-α gene can contribute to severe PE in Taiwanese Han populations. The results of our study are totally different to previous Iranian studies, but have some similarity to a previous UK study. Further studies are required to confirm the roles of rs1800629 and rs361525 in PE with circulating TNF-α in PE.

## 1. Introduction

Preeclampsia (PE) occurs in women pregnant for more than 20 weeks with de novo hypertension and proteinuria, and is a devastating disease in maternal–fetal medicine [1,2]. Maternal complications include the notorious eclampsia, various end-organ damage such as liver and renal injury, and even increased risks of future permanent organ damage or cardiovascular diseases. Common fetal consequences include intrauterine growth restriction, premature birth, and even stillbirth. For pregnant women affected with severe PE, the only cure for this condition is to deliver the placenta, often ending with an indicated preterm birth. Intensive care of the preterm newborn may be needed, with potential long-term sequelae. 

Although there have been extensive studies on the pathogenesis of preeclampsia, the real etiology is still unclear. Immune maladaptation, placental and endothelial dysfunction, abnormal trophoblast differentiation, and exaggerated systemic inflammatory response may be the etiology of PE [3]. Cytokine tumor necrosis factor (TNF)-α may play a key role in the pathogenesis of PE [4,5]. The impaired placental perfusion as a result of defective maternal spiral artery remodeling leads to endothelial activation [4]. In addition, genes that have been found to display altered placental expression in patients with PE include TNF-α, intercellular adhesion molecule 1, Integrin, interferon-γ, and so on [4]. The relationship between cytokines and the development of PE has been extensively studied, and evidence suggests that pro-inflammatory cytokines, including TNF-α, IL-1, and IL-6, play a significant role in the pathogenesis of severe PE [5]. Additionally, blocking these inflammatory factors has been proposed as a potential treatment strategy due to their involvement in development of PE. TNF-α is a particularly promising target for further study and designing treatment [5]. No anti-TNF-α agents have been used clinically, but they hold potential as a future treatment option. Several studies have shown that TNF-α inhibitors can reduce certain physiological changes mediated by TNF-α in PE [5].

At least three gene–disease association studies have presented controversial results. Chen et al. [6] showed that TNF-alpha mRNA expression is significantly elevated in preeclamptic patients compared with two other control groups. In addition, the high expression of TNF-α may be associated with the TNF1 allele, whose frequency is increased in PE [6]. They concluded that their observations are consistent with a major role for TNF-α in mediating endothelial disturbances, and suggest a key role for TNF-α in the development of preeclampsia. They proposed that TNF-α may alter the balance between prostacycline and thromboxane, increasing the vascular resistance in PE.

In contrast, Dizon-Townson et al. reported that the frequency of the TNF gene is not increased in patients with preeclampsia or HELLP syndrome (hemolysis, elevated liver enzymes, and low platelet count) [7]. Despite not finding significant differences in the allele frequency of the TNF-α gene, researchers still believe that TNF-α plays a crucial role in the development of PE. The lack of significant differences in the allele frequency does not negate the potential significance of TNF-α in the pathogenesis of PE [7]. However, Haggerty et al. reported preeclamptic white women were more likely than normotensive white women to carry the up-regulating TNF-α-308 A/A (odds ratio, 4.1; 95% CI, 1.1–15.3) genotype, and concluded that cytokine genotypes were associated with preeclampsia and may identify women who are at high risk for preeclampsia [8]. The researchers concluded that genetic polymorphisms may contribute to excessive immune system stimulation during pregnancy, thus increasing the risk of developing PE, underlying the genetic susceptibility to the disease [8]. In addition, Saarela et al. observed that the polymorphisms of the TNF-α gene showed a significant haplotype association with susceptibility to preeclampsia in the Finnish population [9]. Specifically, the C-A haplotype of the two polymorphisms, C-850T and G-308A, was found to be associated with a higher risk of PE, while the T-G haplotype was less common in women with PE [9]. To date, no one has studied the haplotype association with preeclampsia in a Taiwanese Han population.

Given these contradictory results, the TNF-α gene is a good candidate gene for PE research. To date, at least two polymorphisms of the TNF-α gene have been reported [7,9,10,11]. We conducted this study to investigate the regulatory regions of the TNF genes, by investigating two promoter polymorphisms, TNFA-308G/A (rs1800629) and -238G/A (rs361525), known to influence TNF expression, and their relationship to PE. We use these two polymorphisms as a basis to find nearby new markers and undertake a haplotype study between the TNF-α gene and PE.

## 2. Materials and Methods

### 2.1. Ethics Statement

This is an observational, monocentric, case–control study. The Institutional Review Board (IRB) of National Cheng Kung University Hospital (NCKUH) approved this study (IRB No.: HR-95-43 and HR-100-066). All participants have written informed consent.

### 2.2. Patients

All subjects in this study were from an ethnically Taiwanese Han population. We retrospectively collected 74 cases of severe PE meeting the criteria of the American College of Obstetricians and Gynecologists at the Department of Obstetrics and Gynecology of NCKUH from 1 August 2006 to 31 July 2011 [12,13]. From January 2006 to December 2010, 119 pregnant women who delivered at the Department of Obstetrics and Gynecology of NCKUH and did not have PE served as normal controls. Patients with severe PE were called back for blood tests with informed consent. The above diagnoses were made according to the medical records of NCKUH, which were reviewed carefully to confirm the fulfillment of the diagnostic criteria described below based on medical history, clinical manifestations, and laboratory data. During the period, pregnant women who had given birth after uncomplicated pregnancies and who had at least two normal pregnancies cared for at NCKUH provided blood samples as normal controls. All participants in this study were matched for fetal sex, parity, gestational age, and maternal age to within 5 years. None of the normal controls had clinical signs of PE or other medical or pregnancy complications. At the time of blood sampling, none were in labor. PE was defined as de novo hypertension in a previously normotensive pregnant woman after 20 weeks of gestation with new-onset proteinuria [3,12,13,14,15]. Hypertension was defined as a systolic blood pressure ≥140 mmHg or a diastolic blood pressure ≥90 mmHg on at least two occasions and 4–6 h apart after the 20th week of gestation in previously normotensive women [3,12,13]. Proteinuria was defined as 300 mg or more protein excretion in 24 h urine collection, or protein concentration of 300 mg/L or higher in urine (≥1+ on dipstick) [3,12,13]. Severe PE was defined as more than one of the following: blood pressure of 160/110 mmHg or higher, excretion of 5 g or more of protein in a 24 h urine sample, the presence of multiorgan involvement such as oliguria, pulmonary edema, visual or cerebral disturbance and pain in the epigastric area or right upper quadrant, abnormal liver enzymes, and thrombocytopenia (platelet count < 100,000 per μL) [3,12,13]. Gestational age was based on the precise date of the last menstrual period or/and ultrasound measurement of crown–rump length in the first trimester. Women who fulfilled PE criteria but did not have severe PE were not included in this study. Other exclusion criteria were as follows: fetal abnormalities, autoimmune disorders, diabetes mellitus, chronic hypertension, chorioamnionitis, premature rupture of membranes, or multiple pregnancies. 

### 2.3. Gene Analysis

Blood samples from a peripheral vein were obtained for gene study. We followed the methods described by Saarela et al. to examine the polymorphisms of the TNF-α gene promoter [9]. DNA was extracted from blood using the Puregene System (Gentra Systems, Research Triangle Park, NC, USA) and stored at 4 °C until analyzed. 

The TNF-alpha polymerase chain reaction (PCR) was constructed to identify a polymorphism at G-308A in the promoter region. A total of 500 ng of patient DNA was added to a 50-µL volume containing 10 mM Tris HCl (pH, 8.3), 1.5 mM MgCl_2_, 50 mM KCl, 1.0 mM of each deoxynucleotide triphosphate, 20 pmol of each primer, and 1 U of Taq polymerase. Conditions for PCR were denaturation at 94 °C for 3 min followed by 35 cycles at 94 °C for 15 s, 58 °C for 45 s, and 72 °C for 45 s, followed by a final extension at 72 °C for 6 min. Primer sequences were 5′AGGCAATAGGTTTTGAGGGCCAT and 5′TCCTCCCTGCTCCGATTCCG. The 107-base pair (bp) PCR products were cleaved with *Nco*I (Boehringer Mannheim; Mannheim, Germany), resulting in cleavage of the TNF-alpha allele into two fragments of 78 and 20 bp, whereas the TNF alpha-2 allele (-308G-A) was not cleaved. The PCR products were separated on a Meta Phor 4% agarose gel (FMC; Rockland, ME, USA) stained with ethidium bromide and visualized under ultraviolet light [7,9,10].

For the amplification of the G-238A polymorphism in the promoter region, we used primers of the downstream and upstream primers as follows: upstream primer: 5′-TCC TGC ATC CTG TCT GGA AGT TAG-3′; downstream primer: 5′-TCA CAC TCC CCA TCC TCC CGG CT-3′, which generate a PCR product of 182 base pairs. The PCR was performed in a final volume of 20 µL that contained 200 ng genomic DNA, 20 picomoles of each primer, each of four dNTP at 250 mmol/L, and 2 U *Taq* polymerase (PROTECH). PCR conditions for both polymorphisms were as follows: 94°C for 4 min, 35 cycles at 94°C for 30 s, 60°C for 30 s, and 72 °C for 1 min, followed by a final extension step of 5 min at 72°C. After amplification, 8 µL of the product are digested with 5 U of *Nla*IV (New England Biolabs) at 37°C for 6 hrs [7,9,10].

### 2.4. Statistical Analysis

Genotype and allele frequencies were compared using χ^2^ testing, using the statistical package SPSS (SPSS Inc., Chicago, IL, USA). Yates correction of continuity was used when an observed number was ≤5.

## 3. Results

The baseline characteristics and pregnancy outcomes of 74 cases of severe PE and 119 cases of normal controls are shown in Table 1. There was no difference in women with severe preeclampsia and the normal controls in age, parity, and the gender of newborn babies. The newborns from the women with severe PE showed earlier gestational age, lower Apgar scores, and lower birth weight. Our results indicate that preeclampsia contributes to perinatal morbidity and mortality [3,16,17]. Specifically, women affected with severe PE may experience an indicated preterm birth, resulting in earlier delivery at a gestational age, and, subsequently, lower birth weights. 

Genotype distributions and allele frequencies of the *Nco*I polymorphism at the position -238 in the promoter region of the TNF-α gene among normal controls and women with severe preeclampsia are shown in Table 2. Genotype was determined in the 74 women with severe preeclampsia and 119 normal controls. Overall, three genotypes (A/A, A/G, G/G) and two alleles (A, G) were seen. The frequency of homozygotes (A/A) was 94.6% (70 of 74 women) in women with severe PE, and 96.6% (115 of 119 women) in normal controls. The frequency of homozygotes (G/G) was 0% both in women with severe PE and normal controls. The allele frequency of A alleles was 97.3% (144 of 148 alleles) in women with severe preeclampsia and 98.3% (234 of 238 alleles) in normal controls. The allele frequency of G alleles was 2.7% (4 of 148 alleles) in women with severe preeclampsia and 1.7% (4 of 238 alleles) in normal controls. For the *Nla*IV polymorphism at the position -238 in the promoter region of the TNF-α gene, neither genotype distributions nor allele frequencies showed statistically significant differences between normal controls and the women with severe PE.

Genotype distributions and allele frequencies of the *Nla*IV polymorphism at the position -308 in the promoter region of the TNF-α gene among normal controls and women with severe PE are shown in Table 3. Genotype was determined in the 74 women with severe PE and 119 normal controls. Overall, three genotypes (G/G, A/G, A/A) and two alleles (A, G) were seen. The frequency of homozygotes (G/G) was 81.1% (60 of 74 women) in women with severe PE and 57.1% (68 of 119 women) in normal controls. The frequency of homozygotes (A/A) was 5.4% (4 of 74 women) in women with severe PE and 0.9% (1 of 119 women) in normal controls (*p* < 0.001). The allele frequency of G alleles was 87.8% (130 of 148 alleles) in women with severe PE and 78.2% (186 of 238 alleles) in normal controls. The allele frequency of A alleles was 12.2% (18 of 148 alleles) in women with severe PE and 21.8% (52 of 238 alleles) in normal controls (*p* = 0.016). For the *Nla*IV polymorphism at the position -308 in the promoter region of the TNF-α gene, both genotype distributions and allele frequencies showed statistically significant differences among normal controls and the women with severe PE.

The results of haplotype frequencies for the *Nla* IV and *Nco* I polymorphisms at the promoter region of the TNF-α gene in severe preeclampsia and controls are shown in Table 4. The proportions of A/G, A/A, G/G, and G/A in the women with severe preeclampsia were 85.49%, 11.81%, 2.35% and 0.35%, respectively. The proportions of A/G, A/A, G/G, and A/G in the normal controls were 76.47%, 21.82%, 1.68% and 0%, respectively. There was a significant difference between normal controls and the women with severe PE (Table 4).

## 4. Discussion

In this study, we first found that polymorphism at position -308, but not -238, in the promoter region of the TNF-α gene can contribute to severe PE in Taiwanese Han populations. Similar results have previously been reported in Caucasian, Turkish, Tunisian, and Iranian populations [9,18,19,20,21,22,23]. A study by Pfab et al. of 1480 Caucasians who were genotyped for TNF-α-308G/A found that the A allele was associated with proteinuria using dipstick tests at the third trimester [18]. Nevertheless, no association between the genotypes and blood pressure throughout all trimesters among pregnant women was found [18]. Another study by Pazarbaşi et al. involving 40 women with eclampsia and 113 women with PE demonstrated that the A/A genotype frequency at the -308 position was significantly higher both among eclamptic women and those with PE compared to normotensive women [20]. In addition, the T/T genotype at the -805 position was less frequent in these two groups of patients compared to the control group [20]. Nonetheless, they proposed that the functional association between the gene polymorphisms, cytokine levels and development of PE still needed to be investigated [20]. Mirahmadian et al. found that the frequency of A alleles at the G-308A position was significantly higher in the 160 preeclamptic Iranian women compared to the control group (10.62% vs. 0%) [21]. On the other hand, at the -238 position, the allele G was more frequent in women with PE compared to the control group (60.62% versus 50%) [21]. They suggested that the inconsistent results found in studies examining TNF-α gene polymorphisms and their relationship to the development of PE may be due to ethnic heterogeneity. Despite this, they believe that the role of gene polymorphism remains important in the development of PE. [21]. Similarly, work by Mohajertehran et al. has shown that allele A frequency was 24.1% among the 54 women with PE and 8.0% among normal pregnant women [22]. As for the genotype study, the frequency of homozygote G/G was 51.9% in the PE group and 84.0% in the normal control group. No homozygote A/A was detected [22]. Tavakkol Afshari et al. investigated the -238 and the -308 position, and demonstrated that the A allele frequencies were significantly higher at both positions among the 153 preeclamptic women compared to 150 healthy pregnant women [23]. At the former position, 14.3% of the preeclamptic women had the homozygous genotype (G/G), compared to 62% of the control group. For the -308 location, 52.2% of the affected women had the homozygous genotype (G/G), while 84% of the control group had this genotype [23]. Studies on other cytokine genotypes have been less consistent, highlighting the important role of TNF-α in mediating PE. A meta-analysis study revealed that TNF-α-308G/A polymorphism is susceptible to PE. The A allele of TNF-α-308G/A polymorphism enhances the chance of PE, especially in Caucasian and Iranian primiparae [24]. The results indicate that TNF-α-308G/A gene polymorphism may play important roles in the pathogenesis of severe preeclampsia.

TNF-α polymorphisms, such as -308G/A, -850C/T, -238G/A, are possibly associated with PE [20,23,25]. TNF-α rs1800629, a G to A transition in the promoter at position −308, is associated with the level of TNF-α expression and is the most studied polymorphism [26,27,28]. The polymorphism at the *TNFA* 308 locus lies within the promoter region for the gene for TNF-α and may alter the binding of transcription factors, thus leading to increased TNF-α messenger ribonucleic acid synthesis [29,30,31,32]. For example, the TNF2 allele was found more frequently in patients with septic shock who did not survive (52%) compared to those who did survive (24%), but no correlation was established with TNF-α [29]. Nevertheless, TNF-α production stimulated with lipopolysaccharide has been shown to be correlated with polymorphism at the -308 position of the promotor region [30]. With transition to A, transcriptional activity of TNF-α increases with more p29 production [32]. Some of the results of previous studies on the promoter polymorphisms of the TNF-α gene have been inconsistent and may have shown only a weak association with various diseases of interest. The heterogeneity of the diseases may contribute to the difficulties in establishing correlations between different studies, and multiple genes may be interacting [32]. Despite the critical role of TNF-α in inflammation regulation, the evidence on the impact of TNF-α genotype on diseases and their outcomes has been conflicting. However, the genetic regulation of TNF-α still holds significance and its study is valuable [31]. The association between polymorphisms in the TNF-α gene and various inflammatory conditions, including infections, autoimmune diseases, transplantation, and even cancers, has been described [31]. 

Even though many studies have demonstrated the association between the risk of PE and TNF-α-308G/A polymorphism in various countries, the link between the risk of PE and TNF-α-308G/A polymorphism is still controversial due to different ethnicity. A meta-analysis encompassing 22 studies included 2459 cases of PE suggested the association between the TNF-α-308G/A polymorphism, and subgroup analysis revealed that the association was evident in certain Caucasian and Iranian populations. [24]. Our study suggested that polymorphism at the -308 position in the TNF-α gene promoter in Taiwanese Han populations may also play a crucial role. Single SNP analysis of 1598 women demonstrated that rs1800629 was associated with increased risks of PE (RR = 1.8) [33]. Though several previous studies have found that the A allele of rs1800629 contributed to PE [8,9,21,22,23,33], there are some studies which have found that more G alleles at rs1800629 contributes to PE [6]. For example, a study in the UK involving 14 women with PE showed that the frequency of homozygotes (G/G) was 64.3% (9 of 14 women) compared to that of 16.7% (2 of 12 women) among normal pregnant women [6]. In addition, they found that individuals who were homozygous for TNF1 had higher TNF-α mRNA expression [6]. In this study, we found more G alleles of rs1800629 in PE, which is same as the study from the UK [6]. Further studies are needed to confirm the roles of alleles of rs1800629 in PE.

The TNF −238A allele (rs361525) has also been implicated in a number of autoimmune diseases including rheumatoid arthritis [34], ankylosing spondylitis [35], systemic lupus erythematosus [36], juvenile idiopathic arthritis [37], Graves’ disease [38], and type I diabetes mellitus [39]. It also plays roles in many infectious diseases including influenza A (H1N1) [40], pulmonary tuberculosis [41,42], hepatitis B [43], infective endocarditis [44], sepsis and septic shock [45], and dengue fever [46]. However, the studies about the role of rs361525 in PE are few. Only four studies in Iranian populations have demonstrated regulatory roles of A alleles of rs361525 in PE [21,23,25,47]. Three studies showed higher A allele frequency of rs361525 in PE [21,23,25]. For instance, a case–control study conducted in Iran involving 153 preeclamptic pregnant women and 140 healthy pregnant women analyzed the polymorphism of rs361525 and found that homozygotes (A/A) were more prevalent in the preeclamptic group compared to the control group (90% versus 10%). Furthermore, the frequency of the A allele was also higher in the case group (51.6%) compared to the control group (18%), with a significant difference of *p* < 0.001. Additionally, the study also revealed significant differences in diastolic blood pressure among patients with different genotypes [25]. On the contrary, one recent study, also from Iran, revealed a protective role of A allele of rs361525 in PE [47]. It revealed that the A allele of rs361525 was less frequent among women with PE than among normal pregnant women (1.8% versus 6.1%, *p* = 0.03) [47]. In addition, the same study also showed that the homozygote (G/G) was more frequent at rs1800629 in women with PE than the control group (91.9% versus 83.5%), and the frequency of the A allele was lower in the effected women than the control group (4.5% versus 8.3%) [47]. Combined together, the GA/GG (-308/ -238) genotypes corresponded to a lower risk of PE [47]. They further evaluated the biological effects of the polymorphism at the -308 position and showed that that the G to A allele substitution results in the loss of a transcription factor DNA-binding site [47]. In our study, we did not find the regulatory role of rs361525 in PE. Due to the above conflicting results and our study, further studies to evaluate the roles of rs361525 in PE are required.

There are some limitations to our study. First, we did not study the serum concentration of TNF-α. Circulating TNF-α can prove a functional relationship between polymorphisms, elevated TNF-α, and PE. Because serum samples were obtained at least one month after delivery, we did not measure circulating TNF-α, because it would be unrepresentative of PE levels. Additional studies of TNF-α polymorphisms and circulating TNF-α with early pregnancy serum samples would confirm the relationship between genetic polymorphisms, excessive inflammation, and PE. Second, the number of subjects is limited, which may contribute to the insignificant difference in rs361525. Third, Hardy–Weinberg equilibrium (HWE) for the -308 variant, and the *p* value of the chi square test was 0.007 for the PE and 0.04 for the controls. There is possibility of undefine allele which may not be found in our study. Further study is required. Fourth, further study is required to transfect the two variants of -308 in cell models and test the effects on a reporter gene. This would link genetic observations to physiology.

## 5. Conclusions

In conclusion, we found that the G alleles of rs1800629 contributed to PE and no allele roles of rs361525 were found in PE in a Taiwanese Han population. The results of our study are totally different to several previous Iranian studies [21,23,25,47] with some similarity to an UK study [6]. Further studies are required to confirm the roles of rs1800629 and rs361525 in PE with circulating TNF-α in PE.

## Figures and Tables

**Table 1 biomedicines-11-00862-t001:** Clinical characteristics and pregnancy outcomes of women with severe preeclampsia and normal controls.

		Women with Severe Preeclampsia	Normal Controls	*p* Value
Number of cases (N)		74	119	NS
Maternal age (years)		31.1 ± 5.1	29.9 ±4.4	NS
Primiparous (%)		61.8%	52.5%	NS
Multiparous (%)		38.2%	47.5%	NS
Gestational age (weeks)		34.9 ± 3.4 **	38.6 ± 1.6	*p* < 0.01
Birth weight (g)		2086 ± 850 **	3201± 431	*p* < 0.01
Apgar score at 1 min		6.1 ± 2.3 **	8.7 ± 0.8	*p* < 0.01
Apgar score at 5 min		8.5 ± 1.3 **	9.9 ± 0.4	*p* < 0.01
Baby gender	Female (%)	52.6%	49.2%	NS
	Male (%)	47.4%	50.8%	NS

** *p* < 0.01 vs. normal controls.

**Table 2 biomedicines-11-00862-t002:** Genotype distributions and allele frequencies of the Nla IV polymorphism at the position -238 in the promoter region of the TNF-α gene among women with severe preeclampsia and normal controls.

	Women with Severe Preeclampsia (N = 74)		Normal Controls (N = 119)		*p* Value
	N	%	N	%	
Genotype distributions					
AA	70	94.6%	115	96.6%	
AG	4	5.4%	4	3.4%	0.49
GG	0	0.0%	0	0.0%	
Allele frequencies					
A	144	97.3%	234	98.3%	0.49
G	4	2.7%	4	1.7%	

**Table 3 biomedicines-11-00862-t003:** Genotype distributions and allele frequencies of the NcoI polymorphism at the position -308 in the promoter region of the TNF-α gene among women with severe preeclampsia and normal controls.

	Women with Severe Preeclampsia (N = 74)		Normal Controls (N = 119)		*p* Value
	N	%	N	%	
Genotype distributions					
GG	60	81.1%	68	57.1%	
GA	10	13.5%	50	42.0%	<0.001
AA	4	5.4%	1	0.9%	
Allele frequencies					
G	130	87.8%	186	78.2%	0.02
A	18	12.2%	52	21.8%	

**Table 4 biomedicines-11-00862-t004:** Haplotype frequencies for the Nla IV and Nco I polymorphisms at the promoter region of the TNF-a gene in severe preeclampsia and controls.

		Haplotype Frequencies				☐	
	2N	A g	A-A	G g	G-A	χ^2^	*p*-Value
Women with severe preeclampsia	148	85.49%	11.81%	2.35%	0.35%	8.14	0.04
Controls	238	76.47%	21.85%	1.68%	0.00%		

## Data Availability

Data are available on request.

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
