# Peer review of "Polymorphisms within the Tumor Necrosis Factor-Alpha Gene Is Associated with Preeclampsia in Taiwanese Han Populations"

_biomedicines, 2023, doi:10.3390/biomedicines11030862_

Round 1
Reviewer 1 Report
This study by Lin et al, is a classical analysis of TNF variants as possible predisposing markers to preeclampsia. The authors started their work by the collection of 74 severe preeclampsia samples and 119 control women.
The major conclusion is that the -238 polymorphism is not associated to the disease, while the -308 polymorphism indicates that apparently the A allele is protective.
There is a striking deviation from Hardy-Weinberg equilibrium (HWE) for the -308 variant. I calculated a p value by chi square and for p=0.007 for the preeclampsias and 0.04 for the controls. How are the authors explaining this? Would it be possible that one allele was sometimes not detected and thus improperly classified a heterozygote as homozygote? In the Preeclampsia group, the prediction yields 1.1 AA genotype while 4 were detected.
The other polymorphism is in perfect accordance with HWE (p=0.97 and 0.98, for the two groups).
I don’t understand the sex ratio presented on Table 1, in particular for the controls 119 x .508 is 60.452, far from 60 or 61 whereas one would expect an integer number.
The degree of novelty of the paper is moderate since polymorphism of the TNF alpha promoter have been studied largely in the past in the context of preeclampsia. The -308 polymorphism has been reported as associated in other populations, but true, not in the Chinese population.
I leave to the Editor the decision of the suitability of this paper to be published in Biomedicines, since the results are rather confirmatory.
For me the major limit of this study, but similar to other previous ones is the absence of mechanistic insights. It would be more ambitious to transfect the two variants of -308 in cell models and test the effects on a reporter gene. It would link genetics observation to physiology.
Author Response
This study by Lin et al, is a classical analysis of TNF variants as possible predisposing markers to preeclampsia. The authors started their work by the collection of 74 severe preeclampsia samples and 119 control women.
The major conclusion is that the -238 polymorphism is not associated to the disease, while the -308 polymorphism indicates that apparently the A allele is protective.
- There is a striking deviation from Hardy-Weinberg equilibrium (HWE) for the -308 variant. I calculated a p value by chi square and for p=0.007 for the preeclampsias and 0.04 for the controls. How are the authors explaining this? Would it be possible that one allele was sometimes not detected and thus improperly classified a heterozygote as homozygote? In the Preeclampsia group, the prediction yields 1.1 AA genotype while 4 were detected.
Answer: Thank you very much for your opinion. We totally agree the possibility of undefine allele which may not be found in our study. We add this point to the third study limitation. “Third, Hardy-Weinberg equilibrium (HWE) for the -308 variant, the P value by chi square is 0.007 for the PE and 0.04 for the controls. There is possibility of undefine allele which may not be found in our study. Further study is required.”
- The other polymorphism is in perfect accordance with HWE (p=0.97 and 0.98, for the two groups). I don’t understand the sex ratio presented on Table 1, in particular for the controls 119 x .508 is 60.452, far from 60 or 61 whereas one would expect an integer number.
Answer: Thank you very much for your opinion. The sex ration is the baby sex ratio. We make it clear in new Table 1.
- The degree of novelty of the paper is moderate since polymorphism of the TNF alpha promoter have been studied largely in the past in the context of preeclampsia. The -308 polymorphism has been reported as associated in other populations, but true, not in the Chinese population. I leave to the Editor the decision of the suitability of this paper to be published in Biomedicines, since the results are rather confirmatory. For me the major limit of this study, but similar to other previous ones is the absence of mechanistic insights. It would be more ambitious to transfect the two variants of -308 in cell models and test the effects on a reporter gene. It would link genetics observation to physiology.
Answer: Thank you very much for your opinion. We add the fourth study limitation. “Fourth, transfect the two variants of -308 in cell models and test the effects on a reporter gene will link genetics observation to physiology. Furthers study is required.”
Reviewer 2 Report
Dear Authors,
The work is interesting, but it needs major changes.
1. table 1
- I suggest including in the table (or in the text) how many women in the study group had HELLP syndrome and eclampsia,
- it would be good to add a column with p-values for each row,
- no explanation of what the symbols mean: ** and #
2 There is confusion in the numbering of Tables 2 and 3 in the text, similarly the placement of the tables in relation to the text.
3. line 165- "For the NlaIV Polymorphism at the position -238 in the promoter region of TNF-α gene, both genotype distributions and allele frequencies showed statistically significant differences among normal controls and the women with severe preeclampsia."
Rather, it should read: For the NcoI Polymorphism at position -308...
4. line 192- twice mentioned A/G, should be A/G and G/A
5. line 218, 242 - the statistically significant difference was for A alleles (not G !) - there were more of them in the healthy control group. Misinterpreted the results obtained and hence drew incorrect conclusions. Interestingly, in the abstract of the paper, the conclusions correspond to the research results presented.
6 In view of the above, the discussion and conclusions should be corrected. The higher number of AA homozygotes among women with PE should also be commented on.
Author Response
The work is interesting, but it needs major changes.
- table 1
- I suggest including in the table (or in the text) how many women in the study group had HELLP syndrome and eclampsia,
- it would be good to add a column with p-values for each row,
- no explanation of what the symbols mean: ** and #
Answer: Thank you very much for your opinion. We reorganize Table 1 and add p-value at each row. We delete #. **: P< 0.01 vs. normal controls
2 There is confusion in the numbering of Tables 2 and 3 in the text, similarly the placement of the tables in relation to the text.
Answer: Thank you very much for your opinion. We are sorry for the displacement of the text. It has been corrected to align with the information in the accompanying tables.
- line 165- "For the NlaIV Polymorphism at the position -238 in the promoter region of TNF-α gene, both genotype distributions and allele frequencies showed statistically significant differences among normal controls and the women with severe preeclampsia."
Rather, it should read: For the NcoI Polymorphism at position -308...
Answer: Thank you very much for your opinion. The paragraph has been moved to align with the context and tables, and the name of the polymorphism position has been corrected accordingly.
- line 192- twice mentioned A/G, should be A/G and G/A
Answer: Thank you very much for your opinion. We have made the correction.
- line 218, 242 - the statistically significant difference was for A alleles (not G !) - there were more of them in the healthy control group. Misinterpreted the results obtained and hence drew incorrect conclusions. Interestingly, in the abstract of the paper, the conclusions correspond to the research results presented.
Answer: Thank you very much for your opinion. As discussed, our study on the position -308 in the TNF-α gene promoter region yielded results consistent with the work by Chen et al. (Reference 6), who found higher occurrences of TNF1 (-308G) in preeclamptic women. However, these results contradict some other literature. To further understand the role of position -308 in the TNF-α gene promoter region, we recommend future research.
6 In view of the above, the discussion and conclusions should be corrected. The higher number of AA homozygotes among women with PE should also be commented on.
Answer: Thank you very much for your opinion. We have discussed more about the higher number of AA homozygotes among preeclamptic women in previous studies and compared them to our study results in the discussion section.
Round 2
Reviewer 2 Report
Dear Authors,
Thank you for improving the paper, now it is more apparent. I have no more comments, I accept it in its present form.